# LLMTRACE: A CORPUS FOR CLASSIFICATION AND FINE-GRAINED LOCALIZATION OF AI-WRITTEN TEXT

## ABSTRACT

The widespread use of human-like text from Large Language Models (LLMs) necessitates the development of robust detection systems. However, progress is limited by a critical lack of suitable training data; existing datasets are often generated with outdated models, are predominantly in English, and fail to address the increasingly common scenario of mixed human-AI authorship. Crucially, while some datasets address mixed authorship, none provide the character-level annotations required for the precise localization of AI-generated segments within a text. To address these gaps, we introduce LLMTrace, a new large-scale, bilingual (English and Russian) corpus for AI-generated text detection. Constructed using a diverse range of modern proprietary and open-source LLMs, our dataset is designed to support two key tasks: traditional full-text binary classification (human vs. AI) and the novel task of AI-generated interval detection, facilitated by character-level annotations. We believe LLMTrace will serve as a vital resource for training and evaluating the next generation of more nuanced and practical AI detection models.

## 1 INTRODUCTION

The rapid advancement and widespread adoption of Large Language Models (LLMs) have enabled the generation of human-like text at an unprecedented scale. This text is often so convincing that human evaluators struggle to distinguish it from human writing, with performance frequently hovering around chance levels (Milička et al., 2025; Jakesch et al., 2023). This capability, while offering enormous benefits, presents a significant dual-use challenge: the same models that assist in creative writing can also generate misinformation, compromise academic integrity, and automate malicious communication. The unreliability of human detection underscores the critical need for robust, automated systems to maintain a trustworthy digital ecosystem.

However, the effectiveness of any detection model is fundamentally limited by its training data. A review of the current landscape (Section 2) reveals several critical gaps: existing datasets often rely on outdated LLMs, are predominantly English-centric with a scarcity of resources for languages like Russian, and focus on simple classification, overlooking the increasingly common scenario of mixed human-AI authorship.

Perhaps the most significant gap is the absence of large-scale resources for the task of *localizing* AI-generated segments within these mixed texts. While a few datasets support boundary detection between authorship (Table 2), they are often limited to single transition points or lack the precise, character-level annotations required to train granular, high-resolution detectors.

To address these limitations, we introduce LLMTrace, a new large-scale, bilingual (English and Russian) corpus designed to push the boundaries of AI text detection. Our work makes two primary contributions. First, we provide a comprehensive classification dataset built from a diverse suite of 38 modern LLMs across nine domains, utilizing a wide variety of prompt types to create challenging and realistic examples. Second, and most crucially, we introduce the large-scale detection dataset with precise, character-level annotations for mixed-authorship texts. These mixed examples are created through a robust pipeline involving automated gap-filling in human texts, AI-based continuation of human texts, and meticulous manual editing of AI texts by humans, ensuring a high

degree of complexity. By making this resource publicly available, we aim to facilitate a new wave of research into more nuanced, practical, and robust AI detection models.

**Label:** AI
**Model:** gpt-4.1-2025-04-14
**Domain:** story
**Topic id:** 54e0f..
**Prompt type:** UPDATE
**Text:** The amphitheatre was ablaze with fervor as the noble sons of Argentina did contest the stalwart warriors of Holland in the ...
**Prompt:** Rewrite this story in the style of a 19th-century literary classic...

**Label:** MIXED
**Model:** gemini-2.5-flash
**Domain:** article
**Topic id:** a7374..
**Prompt type:** FILL GAPS
**Text:** **Burning Man was without a doubt the most incredible experience of my life.** This huge temporary city in the dust ... ever-growing amount of stimulation. **The sheer scale and creative energy of Black Rock City was overwhelming in the best possible way.** ...
**Prompt:** Fill in the missing sentences in the text marked as <SENTENCE> ...
**AI char intervals:** ((0, 74), (278, 375))

Figure 1: Example annotations from the LLMTrace English dataset, showing a sample from the classification dataset (left) and the detection dataset (right).

## 2 RELATED WORKS

The robust evaluation of AI-generated text detectors requires diverse and challenging benchmarks. We review existing resources for two key tasks: binary classification (human vs. AI) and the localization of AI-generated content within mixed-authorship texts. A summary of these benchmarks is provided in Tables 1 and 2.

### 2.1 CLASSIFICATION BENCHMARKS

Table 1: Statistics of the texts in the classification datasets.

| Dataset | Dataset Size | # Domains | # Lang | # Gen | Various prompt types |
|---|---|---|---|---|---|
| MAGE (Li et al., 2023) | 447,7k | 7 | 1 | 27 | ✓ |
| MGTBench (He et al., 2024) | 21k | 3 | 1 | 6 | ✗ |
| BUST (Cornelius et al., 2024) | 25,2k | 4 | 1 | 7 | ✓ |
| RAID (Dugan et al., 2024) | 6,2M | 8 | 1 | 11 | ✗ |
| DetectRL (Wu et al., 2024) | 235,2k | 4 | 1 | 4 | ✓ |
| HC3 (Guo et al., 2023) | 125,2k | 1 | 2 | 1 | ✗ |
| MULTITuDE (Macko et al., 2023) | 74,1k | 1 | 11 | 8 | ✗ |
| M4GT-Bench (Wang et al., 2024b) | 152,7k | 8 | 8 | 7 | ✓ |
| MultiSocial (Macko et al., 2024) | 472k | 1 | 22 | 7 | ✗ |
| RuATD (Macko et al., 2024) | 215k | 6 | 1 | 13 | ✓ |
| WETBench (Quaremba et al., 2025) | 101,9k | 1 | 3 | 4 | ✓ |
| Peer Review Detection (Yu et al., 2025) | 789k | 1 | 1 | 5 | ✗ |
| GEDE (Gehring & Paaßen, 2025) | 13,4k | 1 | 1 | 2 | ✓ |
| ESPERANTO (Ayoobi et al., 2024) | 720k | 4 | 1 | 8 | ✓ |
| SHIELD (Ayoobi et al., 2025) | 700k | 7 | 1 | 7 | ✗ |
| Beemo (Artemova et al., 2024) | 19,6k | several | 1 | 10 | ✓ |
| MixSet (Zhang et al., 2024) | 3,6k | 6 | 1 | 2 | ✓ |
| **LLMTrace (ours)** | 589,086 | 9 | 2 | 38 | ✓ |

Foundational benchmarks like **MAGE** (Li et al., 2023), **MGTBench** (He et al., 2024), **BUST** (Cornelius et al., 2024), **RAID** (Dugan et al., 2024), **DetectRL** (Wu et al., 2024), and **HC3** (Guo et al., 2023) have laid important groundwork but are often limited by narrow prompt diversity, a small suite of generator models, or restricted domain coverage. Multilingual resources such as **MULTI-TuDE** (Macko et al., 2023), **M4GT-Bench** (Wang et al., 2024b), and **MultiSocial** (Macko et al., 2024) have expanded language coverage but are frequently confined to specific domains like news or social media.

The **RuATD** dataset (Shamardina et al., 2022) is, to date, the only large-scale benchmark for AI text detection in Russian, covering outputs from 14 generators across several tasks like translation,

paraphrasing, and others. While it spans diverse domains, it remains limited to Russian and does not achieve full domain coverage.

Other datasets focus on more specific challenges but lack broad generalizability. Domain-specific corpora, while valuable, target narrow areas like Wikipedia edits (**WETBench** (Quaremba et al., 2025)), scholarly reviews (**Peer Review Detection** (Yu et al., 2025)), or student essays (**GEDE** (Gehring & Paaßen, 2025)). Similarly, robustness benchmarks like **ESPERANTO** (Ayoobi et al., 2024) and **SHIELD** (Ayoobi et al., 2025) explore adversarial attacks or "humanification" but remain limited in scope. Finally, while datasets like **Beemo** (Artemova et al., 2024) and **MixSet** (Zhang et al., 2024) consider mixed authorship, they do so only in a classification setting and at a smaller scale.

## 2.2 DETECTION BENCHMARKS

Another line of research addresses mixed-authorship detection through boundary identification. **M4GT-Bench** (Wang et al. (2024b)) and **RoFT** (Dugan et al. (2023)) (with its extension **RoFT-chatgpt** (Kushnareva et al. (2023))) model texts where an LLM continues a human-written beginning, yielding a single transition point — realistic for some cases but unable to capture multiple or arbitrary boundaries. **TriBERT** (Zeng et al. (2024)) extends this to hybrid student essays with up to three boundaries, though its domain remains limited to education. **CoAuthor** (Lee et al. (2022)) records human–machine collaborative writing across diverse prompts, however, its edits are at times very minor (e.g., few-word changes) and may introduce noise and complicate systematic evaluation.

Table 2: Statistics of the detection datasets. Our LLMTrace dataset includes human, AI, and mixed texts, with the size of the mixed subset noted in parentheses.

| Dataset | Dataset Size | # Domains | # Lang | # Gen | # Intervals | Human collab |
|---------|-------------|-----------|--------|-------|-------------|--------------|
| M4GT-Bench | 31,9k | 2 | 1 | 5 | 1 | ✗ |
| RoFT | 27,6k | 4 | 1 | 5 | 1 | ✗ |
| RoFT-chatgpt | 6,9k | 4 | 1 | 1 | 1 | ✗ |
| TriBERT | 17,1k | 1 | 1 | 1 | 1-3 | ✗ |
| CoAuthor | 1,4k | 2 | 1 | 1 | arbitrary | ✓ |
| **LLMTrace (ours)** | 79,342 (**27,7k mixed**) | 9 | 2 | 31 | arbitrary | ✓ |

# 3 LLMTRACE: DESIGN PRINCIPLES AND CURATION PIPELINE

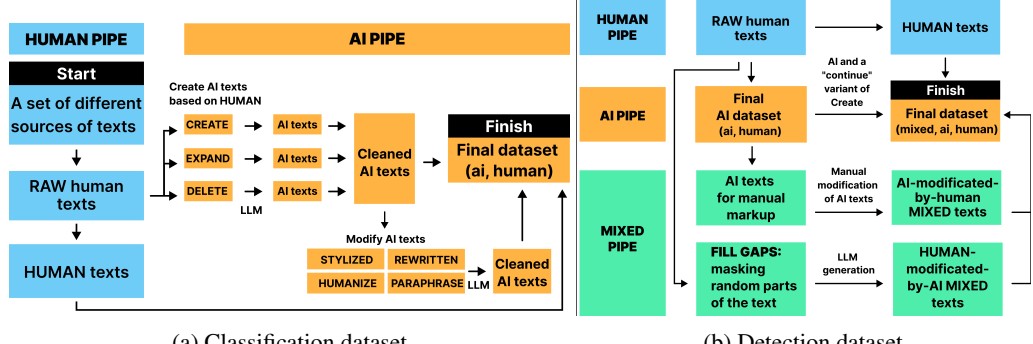

(a) Classification dataset  (b) Detection dataset

Figure 2: The data curation pipelines for the (a) classification and (b) detection datasets. The classification pipeline creates parallel human and AI corpora. The detection pipeline extends this by generating complex mixed-authorship texts through a combination of automated and manual methods.

In this section, we detail the methodology for creating our dataset, LLMTrace, which consists of two large-scale corpora in English and Russian. The dataset is structured to support two primary

tasks: binary classification and segment-level detection, each with specific design principles to ensure diversity, robustness and real-world applicability. The overall pipelines for classification and detection datasets generation are shown in Figure 2.

## 3.1 Target Tasks and Annotation Schema

Our dataset supports two complementary tasks, with examples for both shown in Figure 1.

**Binary Text Classification.** The dataset for this task is formally defined as $\mathcal{D}_{\text{class}} = \{(x_i, y_i)\}_{i=1}^N$, where $x_i$ is the text and $y_i \in \{\text{'human', 'ai'}\}$ is its authorship label. The data is partitioned into standard training, validation, and test sets for the experiments in Section 6.

**AI Interval Detection.** The detection dataset is defined as $\mathcal{D}_{\text{detect}} = \{(x_i, y_i, S_i)\}_{i=1}^M$, where $S_i$ is a set of character-level start/end offsets, $S_i = \{(start_j, end_j)\}$, marking all AI-generated spans. The set $S_i$ is empty for human texts, spans the full text for AI texts, and contains one or more tuples for mixed texts. This dataset is also split into train/valid/test sets for the experiments detailed in Section 6.

## 3.2 Classification Dataset Curation

The curation of our classification dataset (Figure 2a) aimed to create a comprehensive and challenging resource. The design was guided by three core principles: diversity across domains and lengths, a wide range of generator models (including both modern and legacy), and complex generation scenarios designed to produce subtle and sophisticated examples.

**Human Corpus Collection.** The foundation of our dataset is a large, diverse corpus of human-authored texts in English and Russian. To achieve broad domain coverage, we meticulously selected and aggregated data from numerous open-source collections. For the Russian corpus, sources included large-scale collections like SiberianDatasetXL (Denis Petrov, 2023), news and social media corpora from IlyaGusev's datasets[1], and QA datasets such as mfaq (De Bruyn et al., 2021). For the English corpus, we utilized sources such as Common Crawl[2], Wikipedia dumps, news articles (CNN, New York Times), academic abstracts (arXiv, SSRN), and community forums (Reddit, Yelp). A complete list of all data sources is provided in Appendix A. This process resulted in a corpus spanning eight shared domains: **Short-form text** (informal short posts, comments, and messages from social media), **News** (journalistic articles on a wide range of topics), **Question** (texts structured as answers to questions), **Review** (reviews of products, services, or locations), **Factual text** (expository texts containing instructions or factual knowledge), **Poetry** (texts in poetic or verse form), **Story** (fictional narratives, personal stories, and blog posts), and **Article** (general-purpose and encyclopedic articles). For the English dataset, we added a ninth domain, **Paper Abstract**, to specifically cover the scientific writing style prevalent in English-language academia.

**Length-Balanced Sampling.** To ensure structural balance, we sampled human texts uniformly from predefined word-count buckets. This balance was then propagated to the AI corpus by including explicit output-length instructions within our generation prompts. This process ensures the final AI collection mirrors the length distribution of the human corpus, minimizing the risk of models learning to use text length as a simple heuristic for classification.

**AI Text Generation Scenarios.** Using the length-balanced and domain-balanced human corpus as a start, we generated a parallel AI corpus with a diverse library of prompt templates. These templates covered four main scenarios (see Appendix C for examples): generating new text based on a human source (**Create**), shortening it via summarization or simplifying (**Delete**), expanding it with more detail (**Expand**), and modifying existing AI texts through different methods, e.g., humanization or stylization (**Update**). A key benefit of this process is that every AI text is thematically paired with a human text. This design is crucial as it compels detection models to learn subtle stylistic and structural differences, rather than relying on spurious correlations with the topic.

---

[1]https://huggingface.co/IlyaGusev/datasets

[2]https://commoncrawl.org/

**Generator Model Diversity.** To ensure our dataset is not overfitted to the artifacts of a single model family, we utilized a wide spectrum of LLMs. This included: (a) modern proprietary models (e.g., from the Gemini(Comanici et al., 2025) and OpenAI GPT-4(OpenAI, 2023) families); (b) a diverse set of modern open-source models (e.g., Qwen3(Yang et al., 2025), DeepSeek-R1(Guo et al., 2025)); (c) widely-used legacy models (e.g., GPT-3.5(Ouyang et al., 2022)); and (d) for the Russian dataset, language-specific models such as GigaChat(Mamedov et al., 2025) and YaGPT[3]. We intentionally included models of various sizes (from 760M to 72B parameters) and capabilities (with and without advanced reasoning) to capture a broad range of potential generation signatures. A complete list of all used LLMs is provided in Appendix B.

## 3.3 DETECTION DATASET CURATION

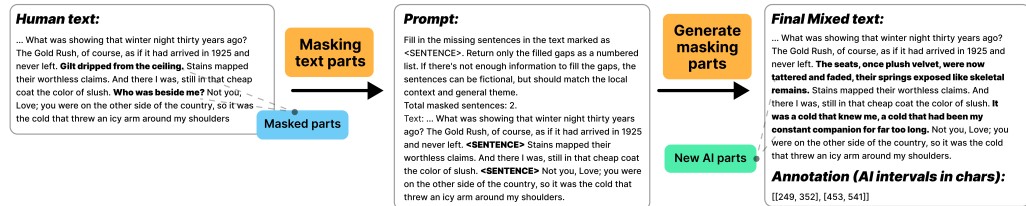

Figure 3: The automated mixed-text generation process: sentences in a human text are masked and then filled in by an LLM, resulting in a text with annotated AI-generated intervals.

The detection dataset was curated to provide realistic and complex examples of human-AI collaboration (Figure 2b). Our design maintained domain and length diversity, allowed for a variable number of AI intervals, and created challenging scenarios through three distinct generation pathways, using our classification dataset as a start.

**Human Text Gap-Filling.** The primary automated method involved masking sentences in human texts and prompting an LLM to fill the gaps coherently, as illustrated in Figure 3. We found this gap-filling task to be highly complex, especially for longer texts with multiple masked sections. Therefore, to ensure the highest quality of generated content, we exclusively used state-of-the-art models (specifically, Gemini-2.5-flash(Comanici et al., 2025) and OpenAI GPT-4, 4o, and o1 families(OpenAI, 2023; Hurst et al., 2024; Jaech et al., 2024)). This process yields a mixed text where the newly generated parts are precisely marked with character-level intervals.

**Human Text Continuation.** Mixed texts were also sourced from our classification dataset's "continue" prompts (a variant of *Create*), where an LLM completes a human-written prefix. These examples provide texts with a single AI-generated interval and crucially inherit the full generator model diversity from the classification set.

**Manual Editing of AI Texts.** Finally, to create the most challenging examples, we tasked a team of editors fluent in both English and Russian with manually modifying AI-generated texts. This process yields a unique sub-corpus where subtle human edits are woven into an AI-generated foundation. Though smaller due to the expensive nature of this process, these examples are vital as the human modifications break consistent AI statistical patterns, forcing models to learn more nuanced and robust features.

## 3.4 POST-PROCESSING AND FILTERING

To ensure data quality, all human and AI texts underwent a rigorous filtering pipeline. This process involved strict language filtering (English or Russian), removing duplicates, low-quality content (e.g., incomplete or repetitive), and any text shorter than five words, including short AI-generated continuations. We also stripped LLM-specific artifacts by filtering common refusal phrases (e.g., "As a language model...") and removing Markdown formatting to normalize the data.

---

[3]https://ya.ru/ai/gpt

# 4 DATASET STATISTICS AND ANALYSIS

This section presents a quantitative analysis of our datasets, focusing on their composition, diversity, and complexity. As detailed in Figure 4, our substantial classification datasets comprise 249k English and 340k Russian samples with a 60/40 AI-to-human ratio. The detection datasets feature a three-way split of human, AI, and mixed texts.

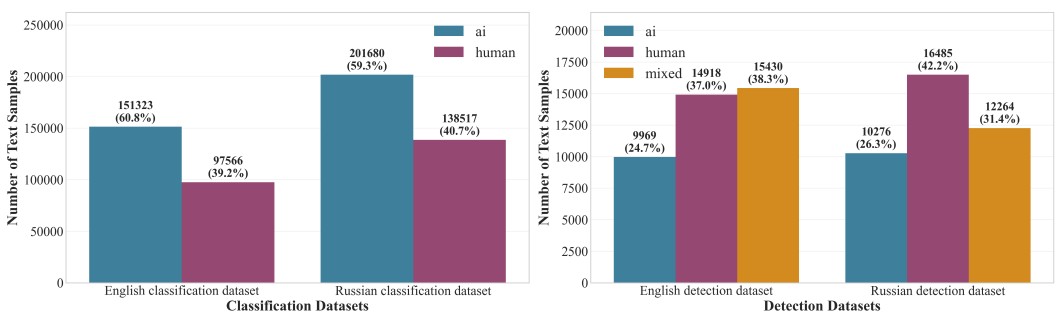

Figure 4: The number of samples for each label in the LLMTrace dataset.

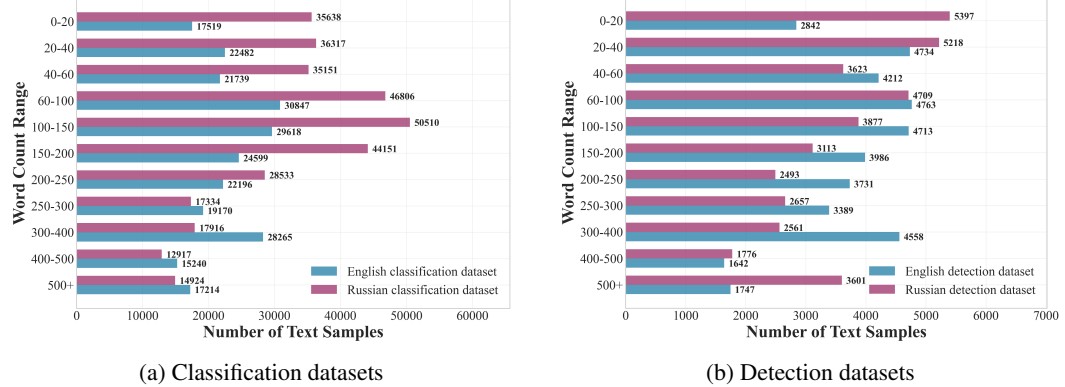

(a) Classification datasets      (b) Detection datasets

Figure 5: The distribution of text samples across different word count ranges for each dataset.

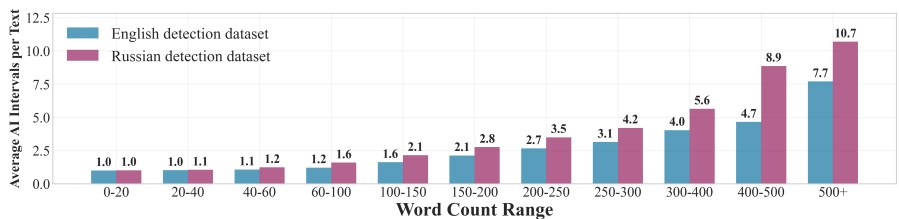

Figure 6: Average number of AI intervals per text in each word count range for the detection datasets.

A key design principle was diversity in both text length and domain. Figure 5 confirms broad coverage across all word count ranges, from short (0-20 words) to long (500+ words) content. The domain coverage is also substantial: for the large-scale classification datasets, each domain is well-represented, averaging approximately 16.8k AI texts for English and 25.2k for Russian. The detection datasets provide robust coverage as well, with each domain containing, on average, over 1.2k AI texts and 1.6k mixed texts for each language. Detailed domain distributions, which are crucial for model generalization, are provided in Appendix D.

Finally, Figure 6 illustrates the structural complexity of our mixed texts. The average number of AI intervals per text correlates positively with word count, increasing for longer texts. This ensures our dataset contains challenging examples with multiple, non-trivial AI insertions.

## 5 QUALITY AND COMPLEXITY ASSESSMENT

Beyond standard statistics, we analyze the quality of our dataset using a dual approach. First, we follow the methodology proposed by Gritsai et al. (2024) to assess the deep structural properties of our generated texts using topological and perturbation-based metrics. Second, we complement this analysis with a suite of classic textual similarity metrics, adapted from the MultiSocial benchmark Macko et al. (2024), to quantify how closely the output of each LLM resembles the source text it was based on. Together, these analyses provide a comprehensive view of how indistinguishable our AI-generated texts are from their human counterparts.

### 5.1 TOPOLOGICAL STATISTICS

Recent work has shown that the intrinsic dimensionality of a text's embedding space can serve as a feature to distinguish between human and machine origins (Tulchinskii et al., 2023). This is often measured using the Persistence Homology Dimension (PHD). While earlier models tended to produce texts with lower PHD than humans, modern LLMs generate text with much more similar topological complexity.

To quantify the similarity between the PHD distributions of human ($h_d$) and AI-generated ($m_d$) texts, we use the symmetric KL-divergence score, $KL_{TTS}$, as proposed by Gritsai et al. (2024):

$$KL_{TTS}(h_d, m_d) = |D_{KL}(h_d||m_d) - D_{KL}(m_d||h_d)|$$

A lower $KL_{TTS}$ score indicates that the two distributions are more similar, suggesting that the AI texts are harder to distinguish from human texts based on this topological feature. Following the methodology of Gritsai et al. (2024), we use a `roberta-base`[4] model to extract text embeddings for PHD calculation.

Our English classification dataset achieves a competitive $KL_{TTS}$ of 0.0189 when benchmarked against other public datasets reported by Gritsai et al. (2024) (see Table 3). Notably, our Russian dataset shows an exceptionally low $KL_{TTS}$ of 0.0032, one of the lowest among all compared Russian-language datasets, demonstrating the high quality of our generation pipeline.

The analysis of the detection datasets in Table 6b provides further insights. For both English and Russian, the $KL_{TTS}$ scores between *AI* and *human* texts remain low, confirming the quality of the purely AI-generated texts in this subset. Most importantly, the score between *mixed* and *human* texts is exceptionally low (0.0017 for EN, 0.0332 for RU). This quantitatively confirms that our methods for creating mixed texts produce challenging examples that are topologically very similar to purely human-written content. This conclusion is further supported by the closely aligned mean PHD values shown in Tables 3 and 6b, especially the near-identical means for human and mixed texts in the English detection dataset. A visual representation of these highly overlapping distributions is provided in Appendix E.

### 5.2 PERTURBATION STATISTICS

This metric assesses dataset quality by measuring how text embeddings shift after minor, meaning-preserving adversarial changes, measured by the $\Delta_{shift}$ score from Gritsai et al. (2024). The perturbation is performed by replacing tokens with synonyms (using WordNet(Miller, 1995) for English and RuWordNet[5] for Russian). The `multilingual-e5-large`[6] encoder is used to generate embeddings. A $\Delta_{shift}$ value close to zero is desirable, as it signifies that both human and AI texts react similarly to the perturbations.

As shown in Table 3, our English classification dataset achieves a near-zero $\Delta_{shift}$ of -0.00317, indicating a very high degree of similarity to human texts in terms of adversarial robustness and

---

[4]https://huggingface.co/FacebookAI/roberta-base

[5]https://pypi.org/project/ruwordnet/

[6]https://huggingface.co/intfloat/multilingual-e5-large

Table 3: Comparison of quality statistics on our classification datasets against other public datasets (values for other datasets are from Gritsai et al. (2024)). Lower scores are better for both $KL_{TTS}$ and $|\Delta_{shift}|$. Our datasets show highly competitive results.

| Dataset | $KL_{TTS} \downarrow$ | $PHD_{human}$ | $PHD_{machine}$ | $\Delta_{shift} \downarrow$ |
|---|---|---|---|---|
| GhostBuster(Verma et al., 2023) | 0.053 | $9.84 \pm 1.18$ | $9.76 \pm 1.15$ | 0.024 |
| MGTBench(He et al., 2024) | 0.043 | $8.77 \pm 1.31$ | $9.97 \pm 1.02$ | 0.031 |
| M4(Wang et al., 2024b) | 0.036 | $7.26 \pm 1.99$ | $8.59 \pm 1.4$ | 0.107 |
| MAGE(Li et al., 2023) | 0.011 | $9.8 \pm 2.14$ | $9.38 \pm 3.04$ | 0.094 |
| SemEval24 Multi(Wang et al., 2024a) | 0.001 | $9.65 \pm 1.81$ | $9.42 \pm 1.44$ | 0.059 |
| RuATD(Shamardina et al., 2022) | 0.007 | $7.33 \pm 1.4$ | $7.46 \pm 1.41$ | 0.315 |
| **LLMTrace (EN)** | **0.0189** | $9.19 \pm 1.87$ | $9.91 \pm 1.64$ | **-0.00317** |
| **LLMTrace (RU)** | **0.0032** | $6.98 \pm 0.82$ | $7.06 \pm 0.69$ | **0.00074** |

Table 4: Topological statistics for our detection datasets. The extremely low $KL_{TTS}$ score between human and mixed texts, especially for English, highlights the complexity of our mixed-authorship examples.

| Dataset | $PHD_{human}$ | $PHD_{mixed}$ | $PHD_{ai}$ | $KL_{TTS} \downarrow$ (mixed vs human) | $KL_{TTS} \downarrow$ (ai vs human) |
|---|---|---|---|---|---|
| **LLMTrace (EN)** | $9.27 \pm 1.87$ | $9.26 \pm 1.83$ | $9.83 \pm 1.65$ | **0.0017** | 0.0233 |
| **LLMTrace (RU)** | $7.12 \pm 0.87$ | $7.15 \pm 0.63$ | $7.27 \pm 0.58$ | **0.0332** | 0.0483 |

outperforming many existing datasets. A visual comparison of the embedding shift distributions for our classification datasets is provided in Appendix F.

## 5.3 TEXTUAL SIMILARITY METRICS

While the preceding metrics assess structural integrity, we also evaluate the direct textual fidelity between AI-generated texts and their human source counterparts. Table 5 summarizes the global average scores for these metrics on our English and Russian classification datasets. We present metrics where higher values indicate greater similarity (METEOR(Banerjee & Lavie, 2005), BERTScore(Zhang et al., 2019), n-gram) and metrics where lower values signify closer resemblance (Levenshtein Distance(LD), LangCheck, MAUVE(Pillutla et al., 2021)). The MAUVE metric is marked with an asterisk (*) to denote that it was calculated on a random sample of 1k text pairs per model. Detailed metric descriptions, as well as per-model and per-prompt type results, are provided in Appendix G.

Table 5: Global average similarity metrics for the English and Russian classification datasets.

| Language | METEOR $\uparrow$ | BERTScore $\uparrow$ | n-gram $\uparrow$ | ED-norm (LD) $\downarrow$ | LangCheck $\downarrow$ | MAUVE* $\downarrow$ |
|---|---|---|---|---|---|---|
| LLMTrace (EN) | 0.2665 | 0.6964 | 0.2164 | 3.0255 | 0.0019 | 0.1713 |
| LLMTrace (RU) | 0.1797 | 0.6767 | 0.1630 | 2.4792 | 0.0045 | 0.1696 |

*MAUVE scores were computed on a random sample of 1k (human, AI) pairs per model. Other metrics were calculated on full data.

## 5.4 DISCUSSION

Our combined quality assessment confirms the high quality of our dataset. The low $KL_{TTS}$ and near-zero $\Delta_{shift}$ scores demonstrate that our generated texts are structurally and topologically indistinguishable from human writing, benchmarking favorably against established datasets. This structural similarity is complemented by high semantic and lexical similarity, as shown in Table 5. For instance, our global BERTScore for English (0.6964) surpasses high-performing models in the MultiSocial benchmark (Macko et al., 2024), indicating strong semantic similarity. Concurrently, our high METEOR (0.2665) and n-gram (0.2164) scores also surpass those reported for the models in their study, confirming substantial lexical overlap. This balance of deep structural similarity and high textual coherence between human and generated texts makes our dataset a challenging and valuable resource for training future detection models.

# 6 EXPERIMENTS

To demonstrate the utility of our dataset, we conduct a series of baseline experiments for both the classification and interval detection tasks. Our primary goal is not to provide an exhaustive comparison of numerous detection models, but rather to establish a single, strong, and reproducible baseline for future research. We adopt the detection methods described in Tolstykh et al. (2024), which utilize a fine-tuned Mistral-7B (Jiang et al., 2023) model for the classification task and a DN-DAB-DETR (Li et al., 2022) model, trained on features extracted from a Mistral-7B-v3[7] model, to localize AI-generated intervals directly at the character level (see hyperparameters in Appendix H).

For the **binary classification task** (human vs. AI), we partitioned our classification dataset into training, validation, and test subsets. We then trained three separate models to evaluate performance in different settings: an English-only model (train/valid/test sizes: *173,511 / 36,949 / 38,429*), a Russian-only model (train/valid/test sizes: *237,929 / 49,747 / 52,521*), and a bilingual model trained on the combined data from both languages. The performance is presented in Table 6a, which reports F1 scores, mean accuracy, and TPR@FPR=0.01. Mean accuracy is the average of the per-class accuracies (human and AI). TPR@FPR=0.01 measures the True Positive Rate for the AI class at a fixed False Positive Rate of 1%. All values are reported in percent (%). A more detailed breakdown of the classification results per domain, text length, and prompt type is available in Appendix I.

For the more challenging task of **localizing AI-generated intervals**, we trained three versions of the DN-DAB-DETR model, corresponding to the English-only, Russian-only, and bilingual settings. The detection dataset was similarly partitioned into dedicated training, validation, and test sets. Specifically, the English subsets contain *27,766 / 5,536 / 7,015* samples, while the Russian subsets contain *26,545 / 5,907 / 6,573* samples for training, validation, and testing, respectively. The results are summarized in Table 6b. We report the standard mean Average Precision (mAP) metric, adapted for one-dimensional intervals. An interval is considered a true positive if its Intersection over Union (IoU) with a ground truth interval exceeds a certain threshold. mAP@0.5 uses a fixed IoU threshold of 0.5, while mAP@0.5:0.95 averages the mAP over multiple IoU thresholds from 0.5 to 0.95.

Table 6: Performance of our baseline models on the classification (a) and detection (b) test sets.

(a) Binary classification results.

| Model | F1 AI | F1 Human | Mean Acc. | TPR@ FPR=0.01 |
|---|---|---|---|---|
| ENG-only | 98.64 | 97.92 | 98.48 | 97.95 |
| RU-only | 98.62 | 98.03 | 98.43 | 97.78 |
| Bilingual | 98.64 | 98.00 | 98.46 | 97.93 |

(b) AI interval detection results.

| Model | mAP@0.5 | mAP@ 0.5:0.95 |
|---|---|---|
| ENG-only | 0.8749 | 0.7555 |
| RU-only | 0.8928 | 0.7839 |
| Bilingual | 0.8976 | 0.7921 |

# 7 CONCLUSION

In this paper, we presented `LLMTrace`, a large-scale, bilingual (English and Russian) dataset designed to address critical gaps in training data for AI text detection, particularly for non-English languages, modern LLMs, and mixed-authorship scenarios.

Our contribution provides resources for two key tasks: binary classification, featuring texts from 38 diverse LLMs across nine domains, and the novel task of AI interval detection with precise, character-level annotations. We demonstrate the dataset's high quality and utility through a comprehensive analysis. A suite of topological, perturbation, and textual similarity metrics confirms that our generated texts are structurally and semantically indistinguishable from human counterparts. Furthermore, strong baseline performance on both tasks validates its immediate suitability for training and evaluating modern detection systems.

By making `LLMTrace` publicly available, we provide a challenging new dataset for the research community. We believe it will support the training and evaluation of the next generation of detection models that are more practical, robust, and capable of handling the nuanced ways AI is used in the real world.

---

[7] https://huggingface.co/mistralai/Mistral-7B-v0.3

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

# A  DATA SOURCES

The tables 7 and 8 list the open-source datasets and public sources used to construct the human-authored corpora for the Russian and English languages.

Table 7: Data sources for the Russian corpus.

| Dataset Name | Source / Link |
| --- | --- |
| SiberianDatasetXL | https://huggingface.co/datasets/SiberiaSoft/SiberianDatasetXL |
| medical_qa_ru_data | https://huggingface.co/datasets/blinoff/medical_qa_ru_data |
| mailruQA-big | https://huggingface.co/datasets/Den4ikAI/mailruQA-big |
| mfaq | https://huggingface.co/datasets/clips/mfaq |
| miracl-ru-corpus | https://huggingface.co/datasets/Cohere/miracl-ru-corpus-22-12 |
| wiki_lingua | https://huggingface.co/datasets/GEM/wiki_lingua |
| taiga | https://huggingface.co/datasets/danasone/taiga |
| IlyaGusev's datasets | https://huggingface.co/IlyaGusev/datasets |
| RussianSuperGLUE | https://github.com/RussianNLP/RussianSuperGLUE |
| xlsum | https://huggingface.co/datasets/csebuetnlp/xlsum |
| mlsum | https://huggingface.co/datasets/reciTAL/mlsum |
| xquad | https://huggingface.co/datasets/google/xquad |
| NestQuad | https://huggingface.co/datasets/NeSTudio/NestQuad |
| sberquad | https://huggingface.co/datasets/kuznetsoffandrey/sberquad |
| ru_sentiment_dataset | https://huggingface.co/datasets/MonoHime/ru_sentiment_dataset |
| restaurants_reviews | https://huggingface.co/datasets/blinoff/restaurants_reviews |
| cedr | https://huggingface.co/datasets/sagteam/cedr_v1 |

Table 8: Data sources for the English corpus.

| Dataset Name / Source | Link / Reference |
| --- | --- |
| medium-articles | https://www.kaggle.com/datasets/fabiochiusano/medium-articles |
| wiki-22-12 | https://huggingface.co/datasets/Cohere/wikipedia-22-12 |
| cnn-dailymail | https://www.kaggle.com/datasets/gowrishankarp/newspaper-text-summarization-cnn-dailymail |
| plot-synopses | https://www.kaggle.com/datasets/cryptexcode/mpst-movie-plot-synopses-with-tags |
| foundation-poems | https://www.kaggle.com/datasets/tgdivy/poetry-foundation-poems |
| amazon-questions | https://www.kaggle.com/datasets/praneshmukhopadhyay/amazon-questionanswer-dataset |
| yahoo-answers | https://www.kaggle.com/datasets/yacharki/yahoo-answers-10-categories-for-nlp-csv |
| amazon-reviews | https://www.kaggle.com/datasets/kritanjalijain/amazon-reviews |
| sentiment-tweets | https://www.kaggle.com/datasets/tariqsays/sentiment-dataset-with-1-million-tweets |
| ask-reddit | https://www.kaggle.com/datasets/gpreda/ask-reddit |
| arxiv-abstracts | https://www.kaggle.com/datasets/spsayakpaul/arxiv-paper-abstracts |
| Common Crawl | https://commoncrawl.org/ |
| SciXGen | Chen et al. (2021) |
| XSum | Narayan et al. (2018) |
| TLDR_news | https://huggingface.co/datasets/JulesBelveze/TLDR$_news$ |
| Reddit WritingPrompts | Fan et al. (2018) |
| ROCStories Corpora | Mostafazadeh et al. (2016) |
| Yelp dataset | Zhang et al. (2015) |
| /r/ChangeMyView (CMV) | Tan et al. (2016) |
| News Outlets | CNN, Washington Post, New York Times |
| Academic Sources | arXiv (CS, physics), Springer's SSRN (HHS) |

# B  USED GENERATION MODELS

The table 9 lists the open-source and proprietary Large Language Models (LLMs) used to construct the AI-written corpora for the Russian and English languages.

Table 9: List of LLMs used for AI text generation.

| Model | Languages | Proprietary | Source |
|---|---|---|---|
| *Open-Source Models* | | | |
| AI21-Jamba-Mini-1.5 | RU | No | https://huggingface.co/ai21labs/AI21-Jamba-Mini-1.5 |
| c4ai-command-r-08-2024 | RU+EN | No | https://huggingface.co/CohereLabs/c4ai-command-r-08-2024 |
| dbrx-instruct | RU+EN | No | https://huggingface.co/databricks/dbrx-instruct |
| DeepSeek-R1-Distill-Qwen-32B | RU+EN | No | https://huggingface.co/deepseek-ai/DeepSeek-R1-Distill-Qwen-32B |
| Falcon3-10B-Instruct | EN | No | https://huggingface.co/tiiuae/Falcon3-10B-Instruct |
| FRED-T5-1.7B | RU | No | https://huggingface.co/ai-forever/FRED-T5-1.7B |
| gemma-1.1-7b-it | RU | No | https://huggingface.co/google/gemma-1.1-7b-it |
| gemma-2-27b-it | RU | No | https://huggingface.co/google/gemma-2-27b-it |
| GLM-4-32B-0414 | EN | No | ://huggingface.co/zai-org/GLM-4-32B-0414 |
| Jamba-v0.1 | RU | No | https://huggingface.co/ai21labs/Jamba-v0.1 |
| Llama-3.1-Nemotron-70B-Instruct-HF | EN | No | https://huggingface.co/nvidia/Llama-3.1-Nemotron-70B-Instruct-HF |
| Llama-3.3-70B-Instruct | RU+EN | No | https://huggingface.co/unsloth/Llama-3.3-70B-Instruct |
| llama-7b | RU | No | https://huggingface.co/baffo32/decapoda-research-llama-7B-hf |
| Magistral-Small-2506 | EN | No | https://huggingface.co/mistralai/Magistral-Small-2506 |
| Meta-Llama-3-8B-Instruct | RU | No | https://huggingface.co/meta-llama/Meta-Llama-3-8B-Instruct |
| Ministral-8B-Instruct-2410 | RU+EN | No | https://huggingface.co/mistralai/Ministral-8B-Instruct-2410 |
| Phi-3-medium-128k-instruct | RU | No | https://huggingface.co/microsoft/Phi-3-medium-128k-instruct |
| Phi-3-mini-128k-instruct | RU | No | https://huggingface.co/microsoft/Phi-3-mini-128k-instruct |
| Qwen2-7B-Instruct | RU | No | https://huggingface.co/Qwen/Qwen2-7B-Instruct |
| Qwen2.5-72B-Instruct | RU+EN | No | https://huggingface.co/Qwen/Qwen2.5-72B-Instruct |
| Qwen3-32B | EN | No | https://huggingface.co/Qwen/Qwen3-32B |
| QwQ-32B | RU | No | https://huggingface.co/Qwen/QwQ-32B |
| ruGPT family (rugpt3large, rugpt2large) | RU | No | Zmitrovich et al. (2023) |
| WizardLM-2-7B | RU | No | https://huggingface.co/dreamgen/WizardLM-2-7B |
| YandexGPT-5-Lite-8B-instruct | RU | No | https://huggingface.co/yandex/YandexGPT-5-Lite-8B-instruct |
| Yi-1.5-34B-Chat | RU | No | https://huggingface.co/01-ai/Yi-1.5-34B-Chat |
| *Proprietary Models* | | | |
| GigaChat series (Pro, Max) | RU | Yes | https://giga.chat/ |
| Google Gemini series (2.0/2.5 flash) | RU+EN | Yes | https://gemini.google.com/ |
| OpenAI GPT series (3.5, 4, 4o, o1, o1-mini, o3) | RU+EN | Yes | https://openai.com/ |
| YaGPT 2 | RU | Yes | https://ya.ru/ai/gpt |

## C   PROMPT GENERATION EXAMPLES

This section provides detailed visual diagrams illustrating the four primary categories of prompt templates used to generate the AI text corpus: *Create*, *Delete*, *Expand*, and *Update*. Each figure demonstrates the end-to-end process, from the selection of a source text to the final AI-generated output.

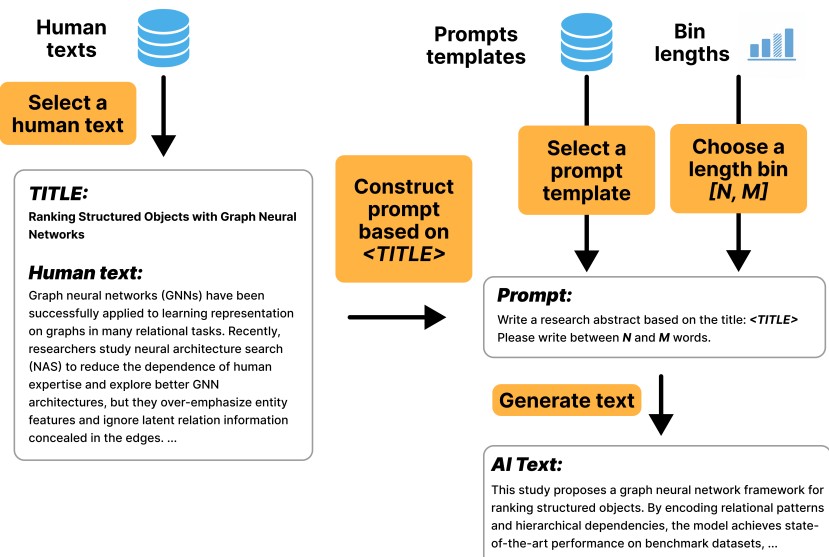

Figure 7: Example of the **Create** generation pipeline, where a research abstract is generated from a title.

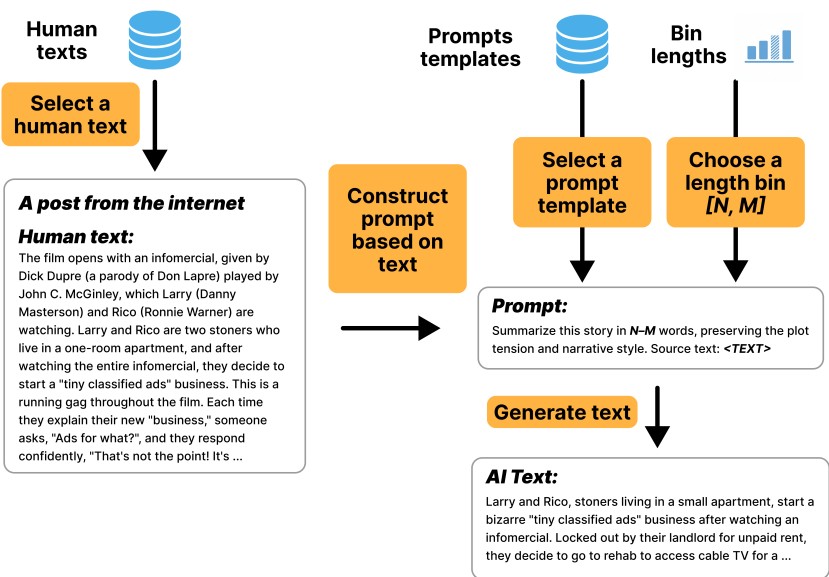

Figure 8: Example of the **Delete** generation pipeline, where a long post is summarized.

The **Create** pipeline (Figure 7) uses a specific attribute from a human text, such as its title, to construct a prompt for generating a new, topically-related AI text within a target length bin.

The **Delete** pipeline (Figure 8) takes an entire human text as input and uses a prompt to generate a condensed, summary version of it, again adhering to a length constraint.

The **Expand** pipeline (Figure 9) operates similarly but with the opposite goal: it uses a prompt to elaborate on a short human text, generating a more detailed and verbose AI text.

Finally, the **Update** pipeline, depicted in Figure 10, is unique. It begins with an already-generated AI text and applies a modification prompt, for instance, to "humanize" the writing style or change its tone, to create a more complex and subtle AI artifact. Note that this pipeline does not typically require a length constraint, as the goal is stylistic transformation rather than content resizing.

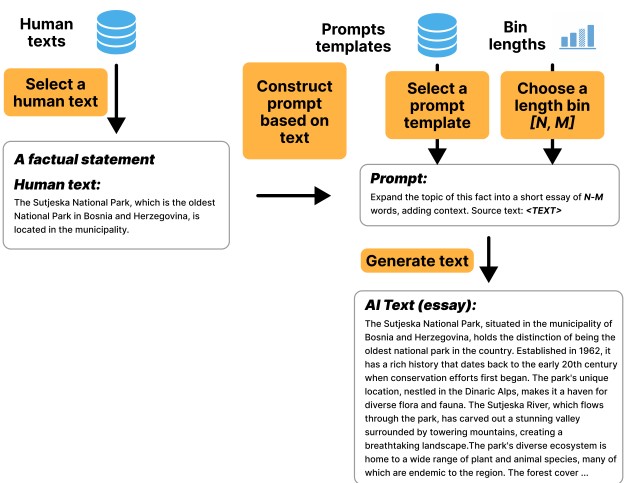

Figure 9: Example of the **Expand** generation pipeline, where a factual statement is expanded into an essay.

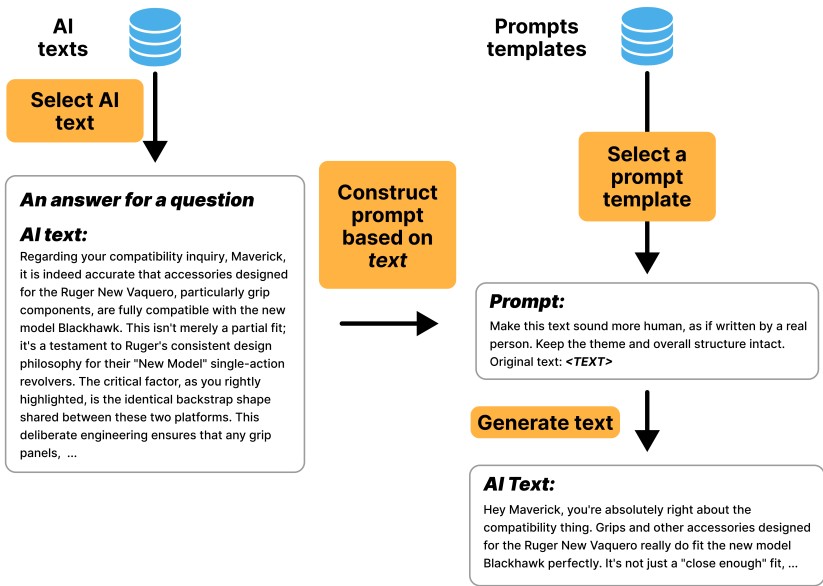

Figure 10: Example of the **Update** generation pipeline, where an AI-generated text is modified to sound more human.

# D  Dataset Statistics and Analysis

This section provides a detailed breakdown of the sample distribution across the various domains for both the classification and detection datasets. Figure 11 illustrates the count of human, AI, and mixed texts within each domain for the English and Russian corpora. These plots visually confirm the broad and balanced coverage of genres, which is a key feature of our dataset designed to enhance the generalizability of detection models.

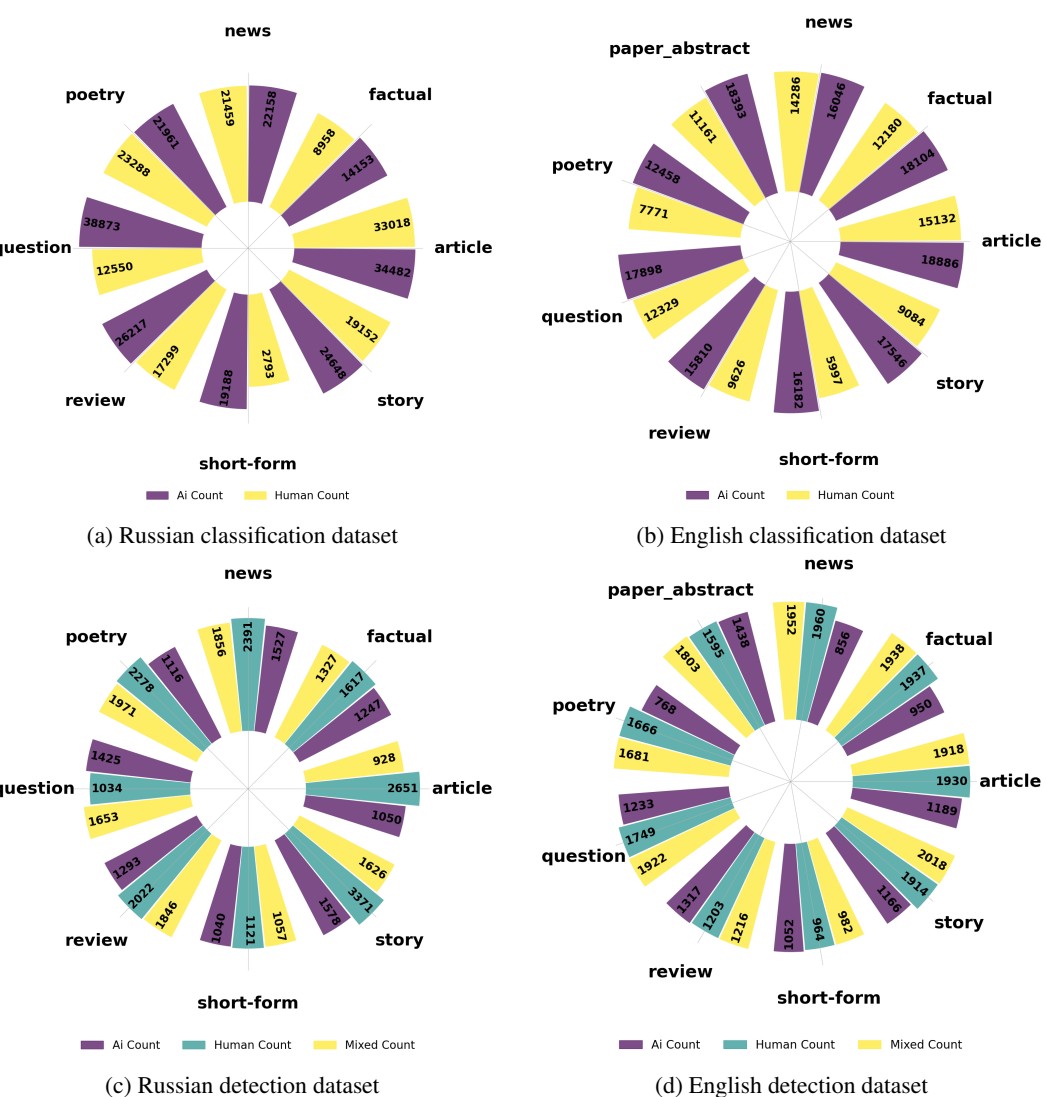

(a) Russian classification dataset  (b) English classification dataset

(c) Russian detection dataset  (d) English detection dataset

Figure 11: Distribution of labels across domains for the classification (a, b) and detection (c, d) datasets. Each plot shows the number of samples for each label type within a specific domain, for both Russian and English languages.

# E  PHD Distributions

Figure 12 provides a visual representation of the Persistence Homology Dimension (PHD) distributions for all categories (human, AI, and mixed) across our classification and detection datasets for both languages. These plots serve as a visual confirmation of the low $KL_{TTS}$ scores and closely aligned mean PHD values reported in Section 5. The significant overlap between the distributions for

all text types highlights the structural and topological similarity between our AI-generated, mixed, and human-authored texts, underscoring the challenge our dataset presents for detection models.

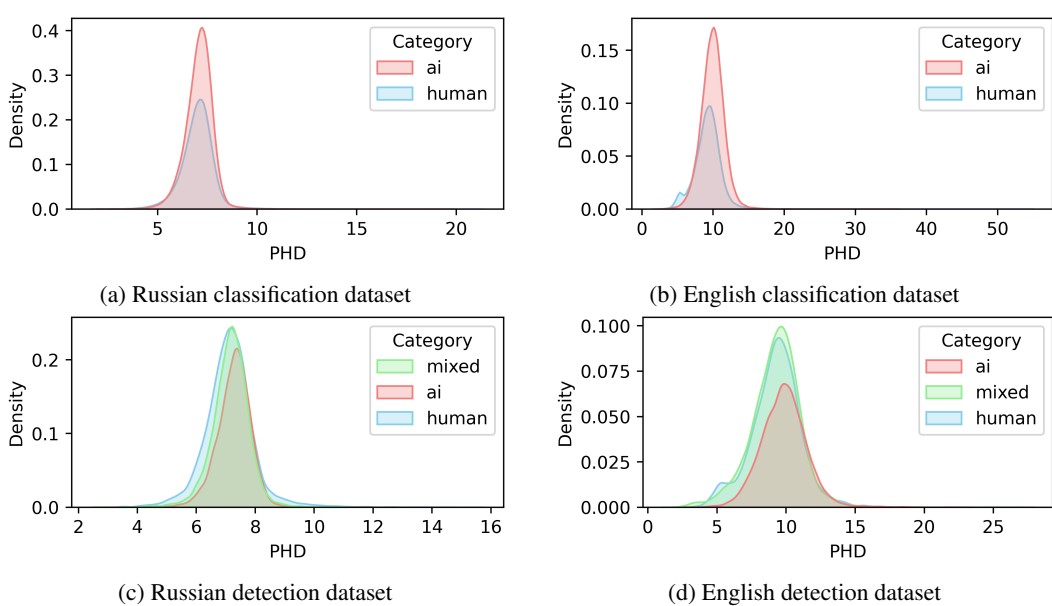

(a) Russian classification dataset

(b) English classification dataset

(c) Russian detection dataset

(d) English detection dataset

Figure 12: PHD comparison for classification (AI vs. Human) and detection (AI vs. Human vs. Mixed) datasets for Russian and English languages.

# F EMBEDDING SHIFT DISTRIBUTIONS AFTER PERTURBATION

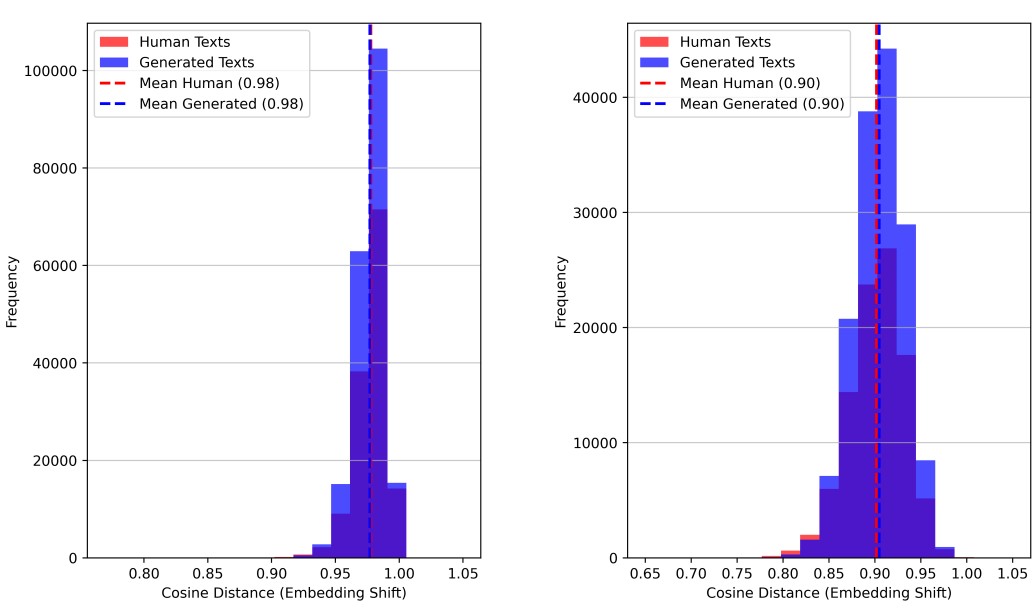

(a) Embedding shifts for the Russian classification dataset.

(b) Embedding shifts for the English classification dataset.

Figure 13: Distributions of cosine distances between original and perturbed text embeddings for human and AI classes. The significant overlap demonstrates similar robustness to synonym-based perturbations for both classes.

This section provides a visual comparison of the embedding shifts for human and AI texts from our classification datasets after undergoing adversarial token perturbation (synonym replacement). The plots in Figure 13 show the distributions of cosine distances between the embeddings of original texts and their perturbed versions.

The close alignment and significant overlap between the distributions for human and AI texts in both the English (Figure 13b) and Russian (Figure 13a) datasets visually confirm the low $|\Delta_{\text{shift}}|$ scores reported in Section 5. This indicates that the AI-generated texts in our dataset exhibit a level of robustness to semantic perturbations that is highly comparable to that of human-authored texts, highlighting the quality and challenge of the dataset.

## G    TEXTUAL SIMILARITY METRIC DESCRIPTIONS AND DETAILED RESULTS

### G.1    SIMILARITY METRIC DESCRIPTIONS

To provide a comprehensive assessment of text similarity, we employ a suite of metrics, each capturing a different aspect of the relationship between the original human text and the machine-generated paraphrase.

- **METEOR** Banerjee & Lavie (2005) (Metric for Evaluation of Translation with Explicit ORdering) is a standard metric in machine translation that measures similarity based on unigram alignments between a reference and a hypothesis text, considering precision, recall, and fragmentation.

- **BERTScore** Zhang et al. (2019) computes similarity by comparing the contextual embeddings of tokens from a reference and a hypothesis text. It uses a multilingual BERT model, making it robust for cross-lingual comparisons and sensitive to semantic similarity beyond exact word matches.

- **n-gram Similarity**[8] is a language-independent metric that calculates the ratio of shared n-grams (in our case, 3-grams) between two strings, providing a measure of surface-level lexical overlap.

- **Levenshtein Distance (LD)**, reported as **ED-norm**, is a character-level metric that counts the minimum number of single-character edits (insertions, deletions, or substitutions) required to change one string into the other. We normalize this value by the character length of the human text. This calculation is performed using the `editdistance`[9] library, consistent with the methodology of the MultiSocial benchmark Macko et al. (2024).

- **LangCheck** measures the percentage of generated texts for which the language differs from the language of the original human text, where both languages are detected using FastText[10]. It serves as an indicator of potential style-imitation artifacts or language inconsistencies.

- **MAUVE** Pillutla et al. (2021) is a distributional metric that measures the gap between the distribution of machine-generated and human-written texts. It uses clustering on text embeddings, obtained from the `google-bert/bert-base-multilingual-cased` model, to quantify the divergence between the two text sets. A lower score indicates that the AI-generated distribution is closer to the human one. Due to its computational intensity, all MAUVE scores were calculated on random data samples. For the per-model analysis (Appendix G.3), our sample consisted of 1,000 pairs per model, for a total of 15,000 English and 26,162 Russian pairs. For the per-prompt type analysis (Appendix G.2), we sampled 3,000 pairs for each of the four prompt types, yielding a total of 12,000 pairs per language.

### G.2    PER-PROMPT TYPE SIMILARITY RESULTS

To further analyze the characteristics of our dataset, we also computed the textual similarity metrics grouped by the type of prompt used for generation. This allows us to understand how different types

---

[8] https://pypi.org/project/ngram/
[9] https://github.com/roy-ht/editdistance
[10] https://fasttext.cc/

of prompts influence the similarity between the AI-generated output and the human source text. The results for the English and Russian datasets are presented in Table 10 and Table 11, respectively.

Table 10: Per-prompt type similarity metrics for the English dataset. Asterisk (*) denotes that the MAUVE metric was calculated on a random sample of 3,000 text pairs per prompt type.

| Prompt Type | METEOR ↑ | BERTScore ↑ | n-gram ↑ | ED-norm ↓ | LangCheck ↓ | MAUVE* ↓ |
|---|---|---|---|---|---|---|
| create | 0.3955 (± 0.3010) | 0.7457 (± 0.1274) | 0.3070 (± 0.2313) | 2.1116 (± 5.0455) | 0.0023 (± 0.0483) | 0.3307 |
| delete | 0.1542 (± 0.1177) | 0.6967 (± 0.0614) | 0.2100 (± 0.1308) | 0.7986 (± 0.1554) | 0.0008 (± 0.0290) | 0.0645 |
| expand | 0.3090 (± 0.1447) | 0.6877 (± 0.0610) | 0.1706 (± 0.1094) | 5.5252 (± 9.2193) | 0.0025 (± 0.0501) | 0.0673 |
| update | 0.2022 (± 0.1045) | 0.6578 (± 0.0526) | 0.1792 (± 0.1017) | 3.4772 (± 7.8540) | 0.0018 (± 0.0426) | 0.1389 |
| **Global** | 0.2665 (± 0.2071) | 0.6964 (± 0.0876) | 0.2164 (± 0.1618) | 3.0255 (± 6.8643) | 0.0019 (± 0.0435) | 0.1675 |

Table 11: Per-prompt type similarity metrics for the Russian dataset. Asterisk (*) denotes that the MAUVE metric was calculated on a random sample of 3,000 text pairs per prompt type.

| Prompt Type | METEOR ↑ | BERTScore ↑ | n-gram ↑ | ED-norm ↓ | LangCheck ↓ | MAUVE* ↓ |
|---|---|---|---|---|---|---|
| create | 0.1616 (± 0.1217) | 0.6568 (± 0.0577) | 0.1307 (± 0.0884) | 2.5010 (± 4.6219) | 0.0037 (± 0.0608) | 0.1719 |
| delete | 0.1235 (± 0.1186) | 0.6894 (± 0.0602) | 0.1684 (± 0.1226) | 1.2259 (± 2.7432) | 0.0060 (± 0.0770) | 0.0829 |
| expand | 0.3223 (± 0.1780) | 0.7053 (± 0.0688) | 0.2179 (± 0.1372) | 4.0885 (± 21.1010) | 0.0068 (± 0.0823) | 0.1175 |
| update | 0.1648 (± 0.1303) | 0.6786 (± 0.0551) | 0.1752 (± 0.1062) | 2.5295 (± 7.7285) | 0.0028 (± 0.0529) | 0.2235 |
| **Global** | 0.1797 (± 0.1482) | 0.6767 (± 0.0623) | 0.1630 (± 0.1135) | 2.4792 (± 9.7793) | 0.0045 (± 0.0670) | 0.1887 |

## G.3 PER-MODEL SIMILARITY RESULTS

The following tables provide a detailed breakdown of performance for each LLM, evaluated separately on the English (Table 12) and Russian (Table 13) datasets.

## H GIGACHECK TRAINING HYPERPARAMETERS

All models were trained using transformers[11] library.

**Classification Model (Mistral-7B)** . The classification model was trained with the following key hyperparameters:

- **Pretrained model:** Mistral-7B-v0.3
- **Sequence length:** max 1024, min 100, random sequence length enabled
- **LoRA:** rank=8, alpha=16
- **Precision:** bf16
- **Batch size per GPU:** 64 (train), 1 (eval)
- **Number of GPUs**: 8
- **Gradient accumulation steps:** 1

---

[11]https://github.com/huggingface/transformers

Table 12: Per-model similarity metrics for the English dataset. Asterisk (*) denotes that the MAUVE metric was calculated on a random sample of 1,000 text pairs per model. Other metrics were calculated on the full English classification dataset.

| Model | METEOR ↑ | BERTScore ↑ | n-gram ↑ | ED-norm ↓ | LangCheck ↓ | MAUVE* ↓ |
|---|---|---|---|---|---|---|
| c4ai-command-r-08-2024 | 0.2515 (± 0.2003) | 0.6903 (± 0.0881) | 0.2119 (± 0.1674) | 2.8851 (± 7.7371) | 0.0017 (± 0.0417) | 0.1765 |
| Qwen2.5-72B-Instruct | 0.2782 (± 0.2083) | 0.7033 (± 0.0849) | 0.2248 (± 0.1661) | 3.0838 (± 6.8450) | 0.0013 (± 0.0360) | 0.3581 |
| Qwen3-32B | 0.2499 (± 0.1859) | 0.6866 (± 0.0816) | 0.2053 (± 0.1462) | 3.2182 (± 6.9384) | 0.0023 (± 0.0479) | 0.1893 |
| dbrx-instruct | 0.2843 (± 0.1877) | 0.7043 (± 0.0828) | 0.2302 (± 0.1591) | 3.0995 (± 6.3418) | 0.0039 (± 0.0625) | 0.2929 |
| DeepSeek-R1-Distill-Qwen-32B | 0.2656 (± 0.1969) | 0.6984 (± 0.0833) | 0.2183 (± 0.1591) | 3.0828 (± 6.5980) | 0.0017 (± 0.0412) | 0.2872 |
| gemini-2.0-flash | 0.2529 (± 0.2212) | 0.6921 (± 0.0930) | 0.2105 (± 0.1652) | 2.7950 (± 6.5796) | 0.0014 (± 0.0379) | 0.3915 |
| gemini-2.5-flash | 0.2644 (± 0.2218) | 0.6878 (± 0.0907) | 0.2064 (± 0.1510) | 3.3609 (± 9.2018) | 0.0016 (± 0.0399) | 0.2379 |
| gpt-4.1-2025-04-14 | 0.2601 (± 0.2041) | 0.6975 (± 0.0876) | 0.2164 (± 0.1485) | 2.7985 (± 5.7855) | 0.0012 (± 0.0351) | 0.2780 |
| Magistral-Small-2507 | 0.3149 (± 0.2476) | 0.7242 (± 0.1006) | 0.2542 (± 0.2036) | 2.5281 (± 5.7650) | 0.0026 (± 0.0509) | 0.4683 |
| Ministral-8B-Instruct-2410 | 0.2831 (± 0.2414) | 0.7135 (± 0.0945) | 0.2329 (± 0.1917) | 2.2827 (± 5.1872) | 0.0018 (± 0.0423) | 0.3413 |
| Llama-3.1-Nemotron-70B-Instruct-HF | 0.2575 (± 0.1717) | 0.6796 (± 0.0768) | 0.2019 (± 0.1356) | 3.7079 (± 7.5270) | 0.0018 (± 0.0425) | 0.1534 |
| o3-2025-04-16 | 0.2257 (± 0.1816) | 0.6798 (± 0.0808) | 0.1870 (± 0.1279) | 3.0114 (± 6.2277) | 0.0022 (± 0.0467) | 0.2437 |
| Falcon3-10B-Instruct | 0.2729 (± 0.2160) | 0.7033 (± 0.0895) | 0.2224 (± 0.1716) | 2.7260 (± 5.7081) | 0.0013 (± 0.0355) | 0.3005 |
| Llama-3.3-70B-Instruct | 0.2611 (± 0.1905) | 0.6895 (± 0.0835) | 0.2107 (± 0.1597) | 3.4868 (± 8.4771) | 0.0017 (± 0.0408) | 0.2358 |
| GLM-4-32B-0414 | 0.2923 (± 0.2193) | 0.7065 (± 0.0865) | 0.2268 (± 0.1653) | 3.0156 (± 5.8937) | 0.0022 (± 0.0464) | 0.3191 |

- **Optimizer:** AdamW

- **Learning rate:** $3 \cdot 10^{-5}$, cosine scheduler with min LR rate scaled by 0.5

- **Warmup steps:** 20

- **Number of epochs:** 20

- **Random seed:** 8888

**Detection Model (DN-DAB-DETR)** . The DN-DAB-DETR detection model was trained with the following key hyperparameters:

- **Feature extractor:** Mistral-7B-v0.3 (frozen)

- **Sequence length:** max 1024, min 100, random sequence length enabled

- **Precision:** bf16 for frozen feature extractor, fp32 for trained DN-DAB-DETR

- **Batch size per GPU:** 64 (train), 1 (eval)

- **Gradient accumulation steps:** 1

- **Number of GPUs**: 8

- **Optimizer:** AdamW

- **Weight decay:** $1 \cdot 10^{-4}$

- **Learning rate:** $2 \cdot 10^{-4}$, cosine scheduler with min LR rate scaled by 0.5

- **Warmup steps:** 100

- **Number of epochs:** 150

- **DETR parameters:** 45 queries, 3 encoder and decoder layers, input embedding dimension is 256

Table 13: Per-model similarity metrics for the Russian dataset. Asterisk (*) denotes that the MAUVE metric was calculated on a random sample of 1,000 text pairs per model. Other metrics were calculated on the full Russian classification dataset.

| Model | METEOR ↑ | BERTScore ↑ | n-gram ↑ | ED-norm ↓ | LangCheck ↓ | MAUVE* ↓ |
|---|---|---|---|---|---|---|
| Yi-1.5-34B-Chat | 0.1957 (± 0.1529) | 0.6961 (± 0.0596) | 0.1965 (± 0.1160) | 1.9419 (± 3.5813) | 0.0065 (± 0.0804) | 0.2268 |
| c4ai-command-r-08-2024 | 0.1616 (± 0.1074) | 0.6787 (± 0.0466) | 0.1776 (± 0.0973) | 2.2883 (± 4.0365) | 0.0014 (± 0.0372) | 0.1627 |
| GigaChat-Max | 0.1497 (± 0.1215) | 0.6781 (± 0.0482) | 0.1693 (± 0.0933) | 2.1745 (± 3.5091) | 0.0010 (± 0.0314) | 0.2129 |
| Jamba-v0.1 | 0.1093 (± 0.1436) | 0.6360 (± 0.0716) | 0.1035 (± 0.1033) | 1.7394 (± 3.0358) | 0.0000 (± 0.0000) | 0.1391 |
| Meta-Llama-3-8B-Instruct | 0.2137 (± 0.1346) | 0.6728 (± 0.0640) | 0.1695 (± 0.0916) | 3.3092 (± 5.8424) | 0.0012 (± 0.0346) | 0.0539 |
| Phi-3-mini-128k-instruct | 0.2120 (± 0.1312) | 0.6461 (± 0.0607) | 0.1456 (± 0.0808) | 3.5791 (± 5.9486) | 0.0055 (± 0.0741) | 0.0513 |
| QwQ-32B | 0.1496 (± 0.1007) | 0.6787 (± 0.0448) | 0.1640 (± 0.0892) | 2.7626 (± 17.3150) | 0.0017 (± 0.0413) | 0.1776 |
| Qwen2.5-72B-Instruct | 0.2117 (± 0.1703) | 0.7000 (± 0.0592) | 0.1956 (± 0.1245) | 2.2224 (± 4.1846) | 0.0014 (± 0.0375) | 0.3678 |
| Qwen2-7B-Instruct | 0.1831 (± 0.1114) | 0.6653 (± 0.0585) | 0.1536 (± 0.0920) | 4.8198 (± 8.1632) | 0.0008 (± 0.0275) | 0.0523 |
| T5 | 0.1775 (± 0.1430) | 0.6979 (± 0.0642) | 0.1388 (± 0.1022) | 2.6308 (± 3.3524) | 0.0000 (± 0.0000) | 0.9973 |
| WizardLM-2-7B | 0.2030 (± 0.1338) | 0.6676 (± 0.0581) | 0.1642 (± 0.0928) | 3.1223 (± 5.5773) | 0.0043 (± 0.0652) | 0.0453 |
| dbrx-instruct | 0.2167 (± 0.1837) | 0.6999 (± 0.0659) | 0.1952 (± 0.1366) | 2.1221 (± 5.2134) | 0.0206 (± 0.1419) | 0.2743 |
| DeepSeek-R1-Distill-Qwen-32B | 0.1732 (± 0.1578) | 0.6879 (± 0.0621) | 0.1738 (± 0.1207) | 3.2682 (± 40.5175) | 0.0223 (± 0.1478) | 0.2339 |
| gemma-1.1-7b-it | 0.1493 (± 0.1038) | 0.6363 (± 0.0542) | 0.1278 (± 0.0718) | 3.9172 (± 6.7532) | 0.0051 (± 0.0714) | 0.0082 |
| GigaCha | 0.1506 (± 0.1314) | 0.6573 (± 0.0663) | 0.1272 (± 0.1007) | 3.2317 (± 5.1930) | 0.0024 (± 0.0492) | 0.1378 |
| gemma-2-27b-it | 0.1730 (± 0.1381) | 0.6810 (± 0.0494) | 0.1724 (± 0.1003) | 1.9631 (± 3.1443) | 0.0015 (± 0.0388) | 0.2404 |
| gpt-3.5 | 0.1766 (± 0.1273) | 0.6646 (± 0.0573) | 0.1474 (± 0.0934) | 3.0603 (± 4.3994) | 0.0015 (± 0.0393) | 0.1169 |
| gpt-4-0125-preview | 0.1076 (± 0.0513) | 0.6201 (± 0.0379) | 0.0990 (± 0.0647) | 5.7928 (± 8.0727) | 0.0047 (± 0.0682) | 0.0846 |
| gpt-4-1106-preview | 0.1053 (± 0.0503) | 0.6182 (± 0.0371) | 0.0934 (± 0.0619) | 5.5213 (± 7.7930) | 0.0021 (± 0.0460) | 0.0629 |
| gpt-4o | 0.1178 (± 0.0795) | 0.6455 (± 0.0463) | 0.0921 (± 0.0570) | 1.0561 (± 0.6252) | 0.0060 (± 0.0771) | 0.1094 |
| llama-7b | 0.2260 (± 0.1361) | 0.6798 (± 0.0577) | 0.1753 (± 0.0953) | 2.7548 (± 4.2883) | 0.0018 (± 0.0422) | 0.3801 |
| Phi-3-medium-128k-instruct | 0.2053 (± 0.1200) | 0.7183 (± 0.0558) | 0.2652 (± 0.1203) | 5.3981 (± 21.8489) | 0.0095 (± 0.0976) | 0.1389 |
| Ministral-8B-Instruct-2410 | 0.2689 (± 0.2380) | 0.7254 (± 0.0799) | 0.2351 (± 0.1815) | 1.8615 (± 8.5790) | 0.0024 (± 0.0485) | 0.3012 |
| o1-mini-2024-09-12 | 0.1881 (± 0.1108) | 0.6748 (± 0.0507) | 0.1849 (± 0.0884) | 1.5011 (± 2.3402) | 0.0000 (± 0.0000) | 0.2687 |
| o1-preview-2024-09-12 | 0.1785 (± 0.1134) | 0.6789 (± 0.0540) | 0.2330 (± 0.1115) | 1.9907 (± 5.2266) | 0.0000 (± 0.0000) | 0.3393 |
| o3-2025-04-16 | 0.1523 (± 0.1003) | 0.6680 (± 0.0464) | 0.1549 (± 0.0849) | 3.0001 (± 8.0777) | 0.0042 (± 0.0649) | 0.1021 |
| ruGPT | 0.2053 (± 0.1250) | 0.6605 (± 0.0595) | 0.1416 (± 0.0901) | 5.8582 (± 9.0776) | 0.0032 (± 0.0566) | 0.2552 |
| Llama-3.3-70B-Instruct | 0.2263 (± 0.1788) | 0.7000 (± 0.0622) | 0.2033 (± 0.1273) | 2.4090 (± 5.8166) | 0.0012 (± 0.0345) | 0.1998 |
| YaGPT | 0.1777 (± 0.1371) | 0.6646 (± 0.0639) | 0.1520 (± 0.0964) | 2.6590 (± 3.7624) | 0.0006 (± 0.0239) | 0.1359 |
| YandexGPT-5-Lite-8B-instruct | 0.2241 (± 0.1791) | 0.7025 (± 0.0630) | 0.2009 (± 0.1269) | 2.0329 (± 3.4634) | 0.0029 (± 0.0536) | 0.2323 |

# I  DETAILED CLASSIFICATION RESULTS

To provide a more granular analysis of our binary classification models' performance, we report the results disaggregated by several key factors. Table 14 shows the performance across different text domains. Table 15 presents the results broken down by text length, grouped into bins based on word count. Finally, Table 16 details the performance for each of the prompt types used during data generation. This detailed breakdown demonstrates the model's robust performance across various conditions and data subsets.

Table 14: Metrics for each dataset and data type.

| Dataset | Data Type | AI F1 | Human F1 | Mean Accuracy | TPR@FPR=0.01 |
|---|---|---|---|---|---|
| English-only | Article | 0,9864 | 0,9829 | 0,9857 | 0,9798 |
| | Factual text | 0,9867 | 0,9803 | 0,9852 | 0,9796 |
| | News | 0,9881 | 0,9866 | 0,9877 | 0,9827 |
| | Paper abstract | 0,9934 | 0,9894 | 0,9923 | 0,9919 |
| | Poetry | 0,9886 | 0,9820 | 0,9863 | 0,9830 |
| | Question | 0,9840 | 0,9775 | 0,9833 | 0,9793 |
| | Review | 0,9778 | 0,9654 | 0,9767 | 0,9629 |
| | Story | 0,9926 | 0,9861 | 0,9910 | 0,9890 |
| | Short-form text | 0,9784 | 0,9439 | 0,9680 | 0,9434 |
| Russian-only | Article | 0,9883 | 0,9878 | 0,9881 | 0,9856 |
| | Factual text | 0,9836 | 0,9743 | 0,9810 | 0,9697 |
| | News | 0,9814 | 0,9805 | 0,9809 | 0,9739 |
| | Poetry | 0,9812 | 0,9825 | 0,9816 | 0,9746 |
| | Question | 0,9921 | 0,9760 | 0,9857 | 0,9849 |
| | Review | 0,9893 | 0,9841 | 0,9878 | 0,9853 |
| | Story | 0,9878 | 0,9846 | 0,9863 | 0,9841 |
| | Short-form text | 0,9784 | 0,8691 | 0,9551 | 0,8720 |
| Bilingual | Article | 0,9890 | 0,9878 | 0,9887 | 0,9854 |
| | Factual text | 0,9853 | 0,9778 | 0,9838 | 0,9772 |
| | News | 0,9844 | 0,9832 | 0,9839 | 0,9787 |
| | Paper abstract | 0,9927 | 0,9882 | 0,9905 | 0,9904 |
| | Poetry | 0,9853 | 0,9839 | 0,9850 | 0,9808 |
| | Question | 0,9888 | 0,9750 | 0,9837 | 0,9815 |
| | Review | 0,9839 | 0,9758 | 0,9826 | 0,9747 |
| | Story | 0,9894 | 0,9845 | 0,9878 | 0,9855 |
| | Short-form text | 0,9789 | 0,9206 | 0,9653 | 0,9056 |

Table 15: Metrics for each dataset and word bin.

| Dataset | # Words | AI F1 | Human F1 | Mean Accuracy | TPR@FPR=0.01 |
|---|---|---|---|---|---|
| English-only | <100 | 0,9747 | 0,9674 | 0,9725 | 0.9500 |
| | 100-400 | 0,9939 | 0,9891 | 0,9936 | 0,9914 |
| | >400 | 0,9860 | 0,9813 | 0,9859 | 0,9778 |
| Russian-only | <100 | 0,9746 | 0,9729 | 0,9740 | 0,9495 |
| | 100-400 | 0,9946 | 0,9860 | 0,9931 | 0,9941 |
| | >400 | 0,9785 | 0,9932 | 0,9804 | 0,9706 |
| Bilingual | <100 | 0,9747 | 0,9710 | 0,9734 | 0,9478 |
| | 100-400 | 0,9946 | 0,9881 | 0,9937 | 0,9935 |
| | >400 | 0,9835 | 0,9882 | 0,9839 | 0,9751 |

Table 16: Metrics for each dataset and prompt type.

| Dataset | Prompt Type | AI F1 | AI Accuracy |
|---|---|---|---|
| English-only | Create | 0,9674 | 0,9368 |
| | Delete | 0,9940 | 0,9880 |
| | Expand | 0,9986 | 0,9972 |
| | Update | 0,9966 | 0,9933 |
| Russian-only | Create | 0,9919 | 0,9839 |
| | Delete | 0,9820 | 0,9646 |
| | Expand | 0,9937 | 0,9874 |
| | Update | 0,9936 | 0,9875 |
| Bilingual | Create | 0,9844 | 0,9692 |
| | Delete | 0,9878 | 0,9759 |
| | Expand | 0,9968 | 0,9936 |
| | Update | 0,9959 | 0,9919 |

Table 17: Metrics for each dataset and generator (only top-5 generators with highest AI F1 metric and top-5 generators with lowest metric are reported).

| Dataset | Prompt Type | AI F1 | AI Accuracy |
|---|---|---|---|
| English-only (**highest metrics**) | llama-3.1-nemotron-70b-instruct-hf | 0,9974 | 0,9949 |
| | qwen/qwen3-32b | 0,9965 | 0,9930 |
| | databricks/dbrx-instruct | 0,9949 | 0,9898 |
| | cohereforai/c4ai-command-r-08-2024 | 0,9943 | 0,9887 |
| | gpt-4.1-2025-04-14 | 0,9933 | 0,9868 |
| English-only (**lowest metrics**) | qwen/qwen2.5-72b-instruct | 0,9891 | 0,9784 |
| | deepseek-ai/deepseek-r1-distill-qwen-32b | 0,9873 | 0,9749 |
| | mistralai/ministral-8b-instruct-2410 | 0,9777 | 0,9565 |
| | zai-org/glm-4-32b-0414 | 0,9766 | 0,9543 |
| | mistralai/magistral-small-2507 | 0,9630 | 0,9565 |
| Russian-only (**highest metrics**) | gemma-1.1-7b-it | 1,0000 | 1,0000 |
| | qwen2-7b-instruct | 0,9986 | 0,9972 |
| | google/gemma-2-27b-it | 0,9980 | 0,9960 |
| | cohereforai/c4ai-command-r-08-2024 | 0,9979 | 0,9959 |
| | phi-3-mini-128k-instruct | 0,9973 | 0,9946 |
| Russian-only (**lowest metrics**) | rugpt | 0,9725 | 0,9465 |
| | llama-7b | 0,9717 | 0,9449 |
| | o1-preview-2024-09-12 | 0,9620 | 0,9268 |
| | jamba-v0.1 | 0,9322 | 0,8730 |
| | microsoft/phi-3-medium-128k-instruct | 0,9091 | 0,8333 |
| Bilingual (**highest metrics**) | gemma-1.1-7b-it | 1,0000 | 1,0000 |
| | o1-mini-2024-09-12 | 1,0000 | 1,0000 |
| | phi-3-mini-128k-instruct | 0,9987 | 0,9973 |
| | qwen/qwq-32b | 0,9964 | 0,9982 |
| | nvidia/llama-3.1-nemotron-70b-instruct-hf | 0,9974 | 0,9949 |
| Bilingual (**lowest metrics**) | o1-preview-2024-09-12 | 0,9268 | 0,9268 |
| | llama-7b | 0,9614 | 0,9256 |
| | mistralai/magistral-small-2507 | 0,9608 | 0,9246 |
| | microsoft/phi-3-medium-128k-instruct | 0,9412 | 0,8889 |
| | jamba-v0.1 | 0,9138 | 0,8413 |

