# OpenReview forum: "LLMTrace: A Corpus for Classification and Fine-Grained Localization of AI-Written Text"
_ICLR.cc/2026/Conference — Submitted to ICLR 2026_

### Official Review · Reviewer_2EPr · 2025-10-29

**Soundness:** 2
**Presentation:** 2
**Contribution:** 2
**Rating:** 2
**Confidence:** 3

**Summary:**

This paper presents LLMTrace, a large-scale detection dataset for more nuanced AI-generated text detection like human-AI mixed texts. Built from 38 modern LLMs across nine domains, two languages (English and Russian), it includes character-level annotations to identify mixed human–AI authorship. Comprehensive quality assessments validate its reliability, making LLMTrace a valuable resource for developing robust and fine-grained detection models.

**Strengths:**

- A large-scale dataset covering 38 modern generators, 9 domains, 2 languages (English and Russian), and 2 detection settings.
- Includes manual edits by human editors applied to AI-generated texts.
- Performs a quality-assessment of the dataset using established metrics from prior work.

**Weaknesses:**

- While the dataset is well-curated and potentially useful, the paper lacks sufficient novelty or insights beyond dataset construction. I encourage the authors to provide deeper analyses or empirical findings that demonstrate new scientific value enabled by this dataset.
- The dataset curation process in this paper (e.g., masking, then filling) doesn’t sound very novel, which is somewhat similar to previous work [1]
- In Section 5, the paper claim that the proposed dataset is more challenging than existing ones, based on quality assessments in previous work. However, in Table 6, the baseline detector already achieves 98% TPR @ 1% FPR, which suggests that the task might not be as challenging as described. The paper should clarify what aspect makes the dataset challenging, or provide evaluation with stronger models to demonstrate this claim more convincingly.

---
References:

[1] Zeng et al. Towards Automatic Boundary Detection for Human-AI Collaborative Hybrid Essay in Education. AAAI 2024.

**Questions:**

See the weaknesses part.

---

### Official Review · Reviewer_Qj9X · 2025-11-01

**Soundness:** 2
**Presentation:** 3
**Contribution:** 2
**Rating:** 2
**Confidence:** 4

**Summary:**

LLMTRACE introduces a large-scale, bilingual (English/Russian) corpus for detecting AI-generated text at two levels.

**Strengths:**

Near-98% F1 with Mistral-7B fine-tuning, but not open source.

**Weaknesses:**

Please check the weakness.

**Questions:**

Add comparison with other mixed-authorship datasets like RoFT, TriBERT, and CoAuthor, which already study boundaries.

What's the core contribution of this paper? In my understanding, it is the detection and its annotation reliability, not the inflated F1s on binary classification.

Mask-and-fill over human sentences and single-prefix continuations will inject regular interval structures. This step will make localization easier than in the wild. Please add corpora from human-in-the-loop editing tools and non-sentence-aligned spans.

Add the details about the human edits over AI text.

Add dual labeling on a sizable subset, span IoU agreement, and an error taxonomy.

Do spans include surrounding function words introduced by AI to connect segments? How are paraphrased human sentences handled when AI “lightly rewrites” them?

Add BiLSTM-CRF, RoBERTa-token classification, and modern encoder-taggers as baselines. And add compute and latency.

It's interesting that near-98% F1 with Mistral-7B fine-tuning. Can you add more details? Length distributions, Unicode punctuation patterns, casing normalization, markdown removal artifacts, and open source code.

 Don't use embeddings from models in PHD/KLTTS and ∆shift analyses, which will risk double-dipping and family bias.

---

### Official Review · Reviewer_ge9R · 2025-11-01

**Soundness:** 3
**Presentation:** 4
**Contribution:** 3
**Rating:** 4
**Confidence:** 4

**Summary:**

The paper introduces LLMTrace, a large-scale, bilingual (English/Russian) corpus for AI-text detection. It targets gaps in prior datasets -- English-centric design, and limited support for mixed authorship and character-level localization -- by providing data built with modern LLMs and fine-grained span labels. The corpus supports two tasks: (1) full-text human-vs-AI classification and (2) interval detection that marks exact start/end offsets of AI-written spans.

**Strengths:**

1. Sound Motivation: Prior AI-detection benchmarks mostly stop at binary detection; this paper tackles the harder, more practical problem of character-level localization in mixed authorship.

2. Good Presentation: The paper is clear, well-structured, and easy to follow.

3. Quality Assurance: The authors support the dataset’s quality with careful analyses.

**Weaknesses:**

1. Assumption on mixed-text: The AI-edited texts are created by filling gaps, i.e., assuming AI polishing is just word/phrase replacement. In reality, it can be much more complex: AI might restructure a paragraph while preserving semantics or make slight changes across all sentences. While some AI edits can be represented with boundaries, this cannot cover all cases. Moreover, restricting mixed text to boundary-based cases may introduce structure-specific artifacts.

2. Narrow experimental coverage: The paper provides a single strong baseline (Mistral-based features + DN-DAB-DETR) rather than broader model comparisons, and its quality checks rely on automated metrics (topological/perturbation/similarity) without human evaluation of span accuracy.

3. Missing some related works: There have been recent works on AI-edited texts, such as [1, 2]. Authors should mention/compare these works/benchmarks.


References:

[1] Almost AI, Almost Human: The Challenge of Detecting AI-Polished Writing

[2] IS CHATGPT INVOLVED IN TEXTS? MEASURE THE POLISH RATIO TO DETECT CHATGPT-GENERATED TEXT

**Questions:**

The figure 2 detection dataset (right side) was a bit confusing to me. It's not very intuitive in its current state. For example, why is there an arrow from 'RAW Human Texts' to 'Final AI Dataset'?

---

### Official Review · Reviewer_2ymN · 2025-11-01

**Soundness:** 2
**Presentation:** 3
**Contribution:** 2
**Rating:** 4
**Confidence:** 4

**Summary:**

This paper provides a new, large-scale benchmark for LLM-generated text detection. The dataset contains both English and Russian texts, includes data from 28 different LLMs, and includes both binary classification as well as boundary detection data

**Strengths:**

1. Overall, this seems like a pretty useful resource for the community. My impression is that there are a lot of LLM detection benchmarks out there, but most of them are either relatively small or low quality. The primary benefit of this benchmark in my opinion is its scale

2. The inclusion of the boundary-detection task is also appreciated, because it is likely a more realistic task than binary classification, and it is also a harder task where there is more remaining headroom (cf. Table 6b)

**Weaknesses:**

1. Even though the paper doesn’t run extensive tests of detection models, it seems that the binary classification benchmark is already almost saturated (Table 6a). The same is true for many other benchmarks in this space, e.g., the RAID benchmark. It seems clear to me that we still don’t have models that can robustly detect AI-generated text, so I would expect to see that reflected in benchmark scores. Additionally, given the high scores of the baseline model, I question whether this benchmark can provide a meaningful ranking over detection methods, at least on the binary classification task

2. Nit: while the topological similarity analysis is interesting, it should be noted that the proposed method doesn’t actually achieve the lowest score for KL_TTS (Table 3), even though it is highlighted as such

3. I find it difficult to contextualize the textual similarity metric results. In particular, no other benchmarks are shown in Table 5, and it seems clear that there has to be *some* difference between the LLM- and human-generated text (or else detection would be impossible)

**Questions:**

1. How does the gap-filling text generation work? Are you randomly masking out some % of sentences and then infilling?

---

### Meta-Review · Area_Chair_niBT · 2026-01-06

**Summary:**

1) Benchmark “difficulty” and saturation concerns (Reviewers 2ymN, 2EPr, Qj9X)

2) Mixed-authorship construction may be unrealistic and bias localization (Reviewers ge9R, Qj9X)

3) Evaluation coverage and baselines are too narrow (Reviewers ge9R, Qj9X)

4) Annotation reliability and validation gaps (Reviewers ge9R, Qj9X)

5) Metrics and analysis clarity and possible methodological issues (Reviewers 2ymN, Qj9X)

6) Related work and positioning (Reviewers ge9R, 2EPr, Qj9X)

7) Presentation / clarity issues (Reviewer ge9R)

8) Legal compliance risk (Reviewer ge9R)

**Reviewer Concerns:**

All reviewers concerns remained as authors have not provided rebuttals

**Reviewer Scores:**

no change as there were no rebuttals

---

### Decision · Program_Chairs · 2026-01-26

Reject